# Regeneration of Osteochondral Defects by Combined Delivery of Synovium-Derived Mesenchymal Stem Cells, TGF-β1 and BMP-4 in Heparin-Conjugated Fibrin Hydrogel

**DOI:** 10.3390/polym14245343

**Published:** 2022-12-07

**Authors:** Madina Sarsenova, Yerik Raimagambetov, Assel Issabekova, Miras Karzhauov, Gulshakhar Kudaibergen, Zhanar Akhmetkarimova, Arman Batpen, Yerlan Ramankulov, Vyacheslav Ogay

**Affiliations:** 1Stem Cell Laboratory, National Center for Biotechnology, Astana 010000, Kazakhstan; 2National Scientific Center of Traumatology and Orthopedics Named after Academician N.D. Batpenov, Astana 010000, Kazakhstan; 3School of Science and Humanities, Nazarbayev University, Astana 010000, Kazakhstan

**Keywords:** osteochondral defect, heparin-conjugated fibrin hydrogel, TGF-β1, BMP-4, controlled release, synovium-derived mesenchymal stem cells, regeneration

## Abstract

The regeneration of cartilage and osteochondral defects remains one of the most challenging clinical problems in orthopedic surgery. Currently, tissue-engineering techniques based on the delivery of appropriate growth factors and mesenchymal stem cells (MSCs) in hydrogel scaffolds are considered as the most promising therapeutic strategy for osteochondral defects regeneration. In this study, we fabricated a heparin-conjugated fibrin (HCF) hydrogel with synovium-derived mesenchymal stem cells (SDMSCs), transforming growth factor-β1 (TGF-β1) and bone morphogenetic protein-4 (BMP-4) to repair osteochondral defects in a rabbit model. An in vitro study showed that HCF hydrogel exhibited good biocompatibility, a slow degradation rate and sustained release of TGF-β1 and BMP-4 over 4 weeks. Macroscopic and histological evaluations revealed that implantation of HCF hydrogel with SDMSCs, TGF-β1 and BMP-4 significantly enhanced the regeneration of hyaline cartilage and the subchondral bone plate in osteochondral defects within 12 weeks compared to hydrogels with SDMSCs or growth factors alone. Thus, these data suggest that combined delivery of SDMSCs with TGF-β1 and BMP-4 in HCF hydrogel may synergistically enhance the therapeutic efficacy of osteochondral defect repair of the knee joints.

## 1. Introduction

Restoration of large osteochondral defects in the knee and hip joints remains one of the most challenging problems in orthopedic surgery because articular cartilage has a relatively avascular structure and limited capacity for self-repair [1,2]. Current methods of managing osteochondral defects include multiple microfractures, osteochondral autograft transfer, osteochondral allograft transplantation, autologous chondrocyte implantation and matrix-assisted autologous chondrocyte implantation [3]. However, these clinical repair methods mainly lead to short-term functional regeneration with the formation of fibrocartilage and cannot provide sustainable restoration of functional hyaline cartilage [4,5]. Therefore, the development of new tissue-engineering approaches, which effectively repair the damaged articular cartilage and subchondral bone plate, is of significant clinical interest for orthopedic surgeons as well as for patients with degenerative/destructive joint disorders.

Currently, tissue-engineering technology using various hydrogel scaffolds, mesenchymal stem cells (MSCs) and growth factors is considered as the most promising therapeutic strategy for osteochondral defect regeneration [6,7,8]. MSCs are multipotent stromal stem cells that possess self-renewal capacity and can differentiate into various specialized cell types such as adipocytes, chondrocytes and osteoblasts [9,10]. MSCs are present in many tissues of human body. The technique of primary culture isolation and in vitro expansion is relatively effortless [11]. Recent studies have revealed that MSCs can be isolated from human and animal synovium [12,13]. One of the advantages of synovium-derived MSC (SDMSC) application is that these cells are tissue-resident stem cells that actively participate in maintaining joint homeostasis and cartilage repair [14,15]. Moreover, it has been demonstrated that SDMSCs have greater proliferation activity and chondrogenic potential in vitro compared to MSCs isolated from bone marrow, periosteum, skeletal muscle and adipose tissue, rendering SDMSCs as an appropriate source for cartilage regeneration [16,17].

During cartilage and bone tissue regeneration, growth factors from transforming growth factor β (TGF-β) superfamily members play a very significant role in the synthesis of extracellular matrix (ECM) and proliferation, as well as participating in chondrogenic and osteogenic differentiation of MSCs [18,19]. Among them, TGF-β1 and bone morphogenetic protein-4 (BMP-4) are important growth factors in regenerative medicine and cartilage tissue engineering [20,21]. It has been shown that TGF-β1 can induce and promote chondrogenic differentiation of MSCs to form ectopic cartilage in vivo [22]. The treatment of MSCs with TGF-β1 can significantly increase the synthesis of type II collagen, aggrecan and deposition of glycosaminoglycans, which are typical for the formation of hyaline cartilage [23]. BMP-4 is another potent signaling molecule involved in the development and maintenance of bone and cartilage [24]. It has been shown that BMP-4 increases the production of aggrecan and type II collagen, induces initial differentiation of MSCs to chondroblasts and facilitates differentiation into mature chondrocytes [25]. It was also demonstrated that genetically engineered MSCs expressing BMP-4 were able to form a hyaline-like cartilage that did not degrade even after 6 months after implantation of these cells into osteochondral defects [26]. These studies have shown that BMP-4 has a high potential for stimulation of chondrogenesis, accelerating the recovery of osteochondral defects and maintaining cartilage structure after regeneration.

However, despite the therapeutic efficacy of recombinant growth factors, the clinical application is still limited due to a short half-life and poor biological activity [27]. Large doses of TGFs and BMPs cause adverse effects, including immune response and abnormal growth of cartilage and bone tissue [28]. Therefore, to overcome the aforementioned limitations, many researchers have developed delivery systems based on natural and synthetic scaffolds for sustained release of growth factors in a target site. Among the scaffolds for growth factors and cell delivery, injectable fibrin hydrogels demonstrated great potential in cartilage and bone tissue engineering. The fibrin hydrogel application is supported by its high water content, biocompatibility, biodegradability, nontoxicity, porous framework, structural similarity to the ECM and the ability to match irregular defects [29,30]. Conjugating heparin to such biodegradable fibrin hydrogel provides biomaterial for the controlled release of heparin-binding proteins such as TGFs and BMPs. Heparin-conjugated hydrogel scaffold can protect growth factors from proteolysis and have the ability to prolong retention of their biological activity in vivo [28]. In addition, it has been shown that heparin can modulate biological activity of TGF-β1, which plays a significant role in cell migration, proliferation, cartilage differentiation and synthesis of cartilage-specific ECM, also involved in suppression of immune response [31,32].

Therefore, considering the advantages of heparin-conjugated hydrogel, SDMSCs, TGF-β1 and BMP-4, we investigated the regeneration of cartilage and subchondral bone tissue in a rabbit osteochondral defect. In this study, we confirmed the hypothesis that co-delivery of SDMSCs with two chondroinductive factors within injectable heparin-conjugated fibrin hydrogel may synergistically promote regeneration of articular cartilage and subchondral bone plate. The proposed approach can promote articular osteochondral defect regeneration and might serve as an effective tool for tissue engineering.

## 2. Materials and Methods

### 2.1. Animals

Male Grey Giant rabbits, 10–12 weeks old, were purchased from the “KletkaMaster” company (Saint Petersburg, Russian Federation). Rabbits were held in large cages (Tecniplast, Buguggiate VA, Italy) at a temperature of 23 °C and relative humidity of 60%. Access to food and water for all experimental animals was ad libitum. All procedures involving laboratory animals and their care were in full compliance with current international laws and policies [33] and were approved by the Local Ethics Committee for Animal Use in the National Center for Biotechnology (Kazakhstan).

### 2.2. Rabbit SDMSCs Isolation and Expansion

Synovial membranes were harvested aseptically from the knee of rabbits under general anesthesia induced by intramuscular ketamine hydrochloride injection at a dose of 25 mg/kg. The synovial membranes were rinsed twice with Dulbecco’s phosphate-buffered saline (DPBS) supplemented with 1% penicillin/streptomycin (Gibco, Grand Island, NY, USA), cut into small tissue fragments (1–2 mm^2^) and dissociated with 0.25% collagenase type II solution (Gibco, Grand Island, NY, USA) in a water bath at 37 °C for 4 h. To discard the remaining tissue fragments, suspension of synovium-derived cells was sieved through the sterile 70 μm cell strainer (Corning, Glendale, AZ, USA). Then, isolated cells were collected by centrifugation, washed twice by DPBS and resuspended in a complete culture medium (CCM) containing α-minimal essential medium (α-MEM), 10% fetal bovine serum (FBS) (Gibco, Grand Island, NY, USA) and 1% penicillin/streptomycin. The cells were plated in a T25 cell culture flask (Corning, Glendale, AZ, USA) and cultured in CCM at 37 °C and 5% CO_2_. Nonadherent cells were removed by day 2 and the adherent cells were cultured before they reached 80–90% confluence. The cells were harvested with TrypLE Express (Gibco, Grand Island, NY, USA) and split in ratio 1:3. The CCM was changed every two days.

### 2.3. Colony-Forming Unit–Fibroblast Assay

Rabbit SDMSCs at passage 4 were detached using TrypLE Express and plated into T25 cell culture flasks at a density of 1000 cells. At day 14, single-cell colonies were rinsed with DPBS and stained with 0.5% crystal violet solution. Then, culture flasks were rinsed three times with DPBS and dried at 37 °C in a thermostat. Stained cell colonies were visualized and counted under a stereomicroscope SZ61 (Olympus, Hamburg, Germany).

### 2.4. Trilineage Differentiation Assay

To assess multilineage differentiation potential, SDMSCs at passage 4 were cultured in adipogenic, chondrogenic and osteogenic differentiation induction media for 21 days, according to the previously described protocol [34]. Adipocytes and osteoblasts were determined by Oil Red O and Alizarin Red S staining, respectively. To evaluate chondrogenic differentiation of SDMSCs, cell pellets were fixed with 10% neutral-buffered formalin (NBF) (Labiko, Saint Petersburg, Russian Federation), paraffin-embedded, sectioned at 5 µm and stained with Toluidine blue.

### 2.5. Preparation of Heparin-Conjugated Fibrin (HCF) Hydrogel

HCF hydrogel was prepared according to the previously described protocol [35]. Briefly, 100 mg of low-molecular-weight heparin (LMWH) (Fisher, Waltham, MA, USA) was dissolved in 100 mL of 0.05 M 2-morpholinoethanesulfonic acid monohydrate. In order to activate the -COOH groups of LMWH, 0.04 mM 1-Hydroxy-2,5-pyrrolidinedione (Sigma, Burlington, MA, USA) and 0.08 mM N-Ethyl-N′-(3-dimethylaminopropyl)carbodiimide hydrochloride (Sigma, Burlington, MA, USA) were added and incubated at 4 °C for 12 h. The solution of activated LMWH was shaken vigorously, precipitated with excess volume of anhydrous acetone (Sigma, Burlington, MA, USA) and lyophilized for 24 h. Subsequently, 100 mg of human plasminogen-free fibrinogen (Sigma, Burlington, MA, USA) was dissolved in 20 mL phosphate-buffered saline (pH 7.4) at 4 °C. This solution subsequently reacted with 60 mg of lyophilized LMWH for 3 h at 4 °C. Following precipitation and lyophilization in similar conditions, powder of heparin-conjugated fibrinogen was dissolved in DPBS. To remove residual LMWH, heparin-conjugated fibrinogen was dialyzed with a dialysis sack (12,000–14,000 Da) (Sigma, Burlington, MA, USA) at 4 °C for 24 h and lyophilized for 48 h to produce fibrinogen conjugated to purified heparin. HCF hydrogel was prepared by mixing heparin-conjugated fibrinogen, (40 mg/mL), human plasminogen-free fibrinogen (60 mg/mL), aprotinin (100 KIU/mL) (Sigma, Burlington, MA, USA), human thrombin (500 IU/mg) (Sigma, Burlington, MA, USA) and calcium chloride (6 mg/mL) (Sigma, Burlington, MA, USA). All components of HCF hydrogel were dissolved in DPBS and sterilized through the polyethersulfone membrane filter with pore size 0.45 µm (TPP, Trasadingen, Switzerland).

Normal fibrin hydrogel for the comparison of in vitro growth factor release and degradation behavior was prepared by dissolving and mixing human plasminogen-free fibrinogen (100 mg/mL), aprotinin (100 KIU/mL), human thrombin (500 IU/mg) and calcium chloride (6 mg/mL) (Sigma, Burlington, MA, USA) in DPBS.

### 2.6. Nuclear Magnetic Resonance (NMR) Spectroscopy

We analyzed heparin-conjugated fibrinogen obtained after lyophilization using a JNN-ECA-500 NMR spectrometer (JEOL, Tokio, Japan) and MestReNova software (version 14.2.0, Santiago de Compostela, Spain). The spectrometer has a usage frequency of 500 MHz, the measurement was performed at a frequency of 500 MHz and using D_2_O at room temperature. Shifts of chemical signals of residual protons or atoms were measured relative to carbon D_2_O water.

### 2.7. In Vitro TGF-β1 and BMP-4 Release Kinetics

The kinetics of TGF-β1 and BMP-4 release from HCF hydrogels was determined with ELISA Quantification Kits (Sigma, Burlington, MA, USA) according to the instructions of manufacturer. We incubated hydrogels containing TGF-β1 (200 ng/mL) or BMP-4 (200 ng/mL) in a 1 mL DPBS at 37 °C under orbital shaking at 50 rpm for 14 days. Fibrin hydrogels containing the same concentration of growth factors were used as controls. The supernatants were collected every 24 h, and the gels were replaced with fresh DPBS. Evaluation of released TGF-β1 and BMP-4 from hydrogels was performed with BioRad 680 plate reader (Biorad, Steenvoorde, France) at 450 nm and 540 nm.

### 2.8. Enzyme-Mediated Degradation Test

To examine the hydrogel degradation kinetics, 100 µL hydrogels were prepared as drops in 35 mm Petri dishes. After gelling, the initial weight of the hydrogels (*w*0) was measured on a precision balance. Next, 2 mL of DPBS containing lysozyme (1 mg/mL; Sigma, Burlington, MA, USA) was added to each Petri dish. The hydrogels were incubated at 37 °C for two weeks. At different time points (day 2, 4, 6, 8, 10, 12 and 14), DPBS containing lysozyme was completely removed from Petri dishes and the final weights of the hydrogels (*wt*) were measured again. The calculations of the remaining hydrogel weights were based on the following: (%) = *wt*/*w*0 × 100.

### 2.9. Cell-Mediated Degradation Test

To examine cell-mediated degradation, the hydrogels were prepared with the final concentration of 2 × 10^5^ SDMSCs/mL and added into wells of 24-well tissue culture plate. The hydrogels were incubated in α-MEM supplemented with 10% FBS and 1% penicillin/streptomycin at 37 °C and 5% CO_2_ for two weeks. Culture medium was replaced every three days. The final weight of the hydrogels was measured and imaged at different time points (2, 4, 6, 8, 10, 12 and 14 d). The remaining hydrogel weights were calculated as above.

### 2.10. Cell Viability and Proliferation Assay

Cell viability of rabbit SDMSCs encapsulated in HCF hydrogel was evaluated with LIVE/DEAD Viability/Cytotoxicity Kit (Thermo Fisher Scientific, Burlington, MA, USA). At days 1, 3 and 7 of culturing, the hydrogels with SDMSCs were rinsed twice with DPBS to remove medium and immersed in staining solution of calcein AM and ethidium homodimer-1 for 1 h at 37 °C. Live and dead cells were detected with Cell Observer microscope (Carl Zeiss, Oberkochen, Germany) at excitation wavelength of 488 and 568 nm. For quantitative analysis, live (green) and dead cells (red) were counted using the ImagePro^®^ Plus software (Media Cybernetics, Rockville, MD, USA). Proliferation of SDMSCs was evaluated using DNA content assay. The amount of DNA in the cells encapsulated in the HCF hydrogel was determined using Quant-iT™ PicoGreen^®^ dsDNA assay kit (Thermo Fisher Scientific, USA) according to the instructions of manufacturer. The fluorescence intensity in the samples was measured at 480 nm excitation and 520 nm emission using the SpectraMax M5 multi-mode microplate reader (Molecular Devices, San Jose, CA, USA).

### 2.11. Osteochondral Defect Model and Implantation of Hydrogels

The rabbits were anesthetized with 25 mg/kg of ketamine hydrochloride intramuscularly. For a local anesthesia, the rabbits received injections of 2% lidocaine hydrochloride subcutaneously in the knee joint area. Hind limbs were fixed and prepared in aseptic conditions for the procedure. The surgery stages in an experimental animal are presented in Appendix A. The osteochondral defects (4 mm in diameter, 3 mm in depth) were created in the patellar groove of the distal femur of the knee joint with a hand-operated drill (Johnson & Johnson DePuy Mitek, Raynham, MA, USA). Before the HCF hydrogel administration, the osteochondral defects were thoroughly washed with sterile DPBS and dried using sterile gauze. Table 1 represents four experimental groups in which the rabbits were randomized for hydrogel implantation.

In the in vivo experiment, the defects were managed using one of the following methods: administration of 70 µL of HCF without growth factors and MSCs (negative control group i, *n* = 8); administration of 70 µL of HCF hydrogel with autologous SDMSCs (group ii, *n* = 8); administration of 70 µL of HCF hydrogel with TGF-β1 and BMP-4 (group iii, *n* = 8); administration of 70 µL of HCF hydrogel with autologous SDMSCs, TGF-β1 and BMP-4 (group iv, *n* = 8). In this study, single-hydrogel implantation served as a baseline for determining the effectiveness of the treatment with growth factors and MSCs separately or in combination; thus, it was not critical to use untreated control group. Hydrogels were implanted to the osteochondral defects using automatic micropipette (Eppendorf, Hamburg, Germany). After gelation of HCF hydrogel within 3 min, the wound was sutured and treated by povidone-iodine. To prevent postoperative complication, the rabbits received intramuscular injection of 3 mg/kg gentamycin for three days postsurgery. After the hydrogel implantation, the rabbits moved in the cages without any restrictions and fully stepped on paws by entire weight.

### 2.12. Macroscopic Evaluation of Cartilage Defects

At week twelve after hydrogel implantation, knee samples were harvested from sacrificed rabbits. Regeneration of the osteochondral defects was evaluated using the International Cartilage Repair Society (ICRS) macroscopic scoring system, which contains three categories: degree of repair, integration to border zone and macroscopic appearance [36].

### 2.13. Tissue Processing and Histological Scoring

The knee joints with osteochondral defects were isolated and fixed in 10% neutral buffered formalin. For decalcification, the tissue samples were placed in electrolyte decalcifying solution (BioVitrum, Saint-Petersburg, Russia) for three days. Next, the samples were sequentially dehydrated in 70%, 95%, 95%, 100%, 100% ethyl alcohol and immersed in xylene. Then, the samples were infiltrated with paraffin, embedded into paraffin blocks and cut into 5 μm sections. Before staining, sections were treated with xylene sequentially rehydrated in 100%, 100%, 95%, 95% and 70% ethyl alcohol and distilled water in order to remove paraffin. The sections from each defect were stained with modified Mayer’s hematoxylin and eosin (H&E), Safranin O/Fast Green, sequentially dehydrated and cleared with ethyl alcohol and xylene and mounted in histological medium Bio Mount HM (Bio-Optica, Milano, Italy).

Collagen type II was detected by immunofluorescence staining. Deparaffinized sections were treated by Proteinase K for 15 min at 37 °C for antigen retrieval. The sections were then washed in TBS/0.025% Triton X-100, blocked with 10% normal goat serum and incubated with primary anti-collagen II antibody (dilution 1:100) (ab34712, Abcam, Cambridge, UK) in humidified chamber at 4 °C overnight. The sections were washed with TBS/0.025% Triton X-100 and incubated with secondary antibody (dilution 1:1000) (A-11012, goat anti-rabbit IgG Alexa Fluor 594, Thermo Fisher Scientific, Burlington, MA, USA) for 45 min at 37 °C. After washing in TBS/0.025% Triton X-100, the sections were mounted in Prolong^TM^ Gold antifade mountant with DAPI (Thermo Fisher Scientific, USA) and visualized with Axio Scope A1 upright microscope (Carl Zeiss, Germany). All acquired images were processed using Image J software (version 1.53t, NIH, Bethesda, MD, USA).

Histological sections after H&E and Safranin O staining (total of 8 images per group) were blindly scored by three independent evaluators based on the previously established histological scoring system for osteochondral defect repair [37]. Histological scoring scale had a maximum score of 28 and a minimum score of 0.

### 2.14. Statistical Analysis

All obtained data are presented as mean ± SD. The statistical significance was calculated using one-way ANOVA followed by Bonferroni’s multiple comparison tests. *p* < 0.05 was considered as statistically significant. Statistical analysis was conducted with software Statistica 6.0 (StatSoft, Tulsa, OK, USA).

## 3. Results

### 3.1. Synthesis of Heparin-Conjugated Fibrin Hydrogel

To prepare HCF hydrogel for co-delivery of TGF-β1 and BMP-4, we first synthesized heparin-conjugated fibrinogen using the method of carbodiimide chemistry, as previously described by Yang et al. [35]. Figure 1 shows activation and conjugation of LMWH with human plasma fibrinogen, which was investigated using high-resolution ^1^H-NMR spectroscopy.

Analysis of the compound by the ^1^H NMR spectrum revealed the presence of both carbohydrate fragment protons and aromatic ring protons. The hydroxyl group of the of the pyranose ring proton signals of the heparin-NHS were detected as singlets in the region of 3.59–4.20 ppm. The chemical shifts of the cyclic imide are recorded as a singlet in the range of 1.86–2.28 ppm and a doublet of 2.73 ppm.

Analysis of the synthesized heparin-conjugated fibrinogen showed that the NH proton signal was recorded as a singlet in the high-field region at 1.03 ppm confirmed by ^1^H NMR spectra. Interestingly, protons 1 and 2 in the succinimide ring at 2.73 ppm were significantly reduced after conjugation. It indicates that the succinimide ester ring is hydrolyzed after the reaction and this leaving group may act as the base of the conjugate to deprotonate the α-position of the protons in the pyranose ring.

Thus, the 1H-NMR spectra showed that the synthesized heparin-conjugated fibrinogen exhibited strong covalent bonds formed between the molecules of activated LMWH and fibrinogen, which makes it suitable for the preparation of HCF hydrogel.

In order to prepare HCF hydrogel, we dissolved heparin-conjugated fibrinogen (40 mg/mL), human plasminogen-free fibrinogen (60 mg/mL) and aprotinin (100 KIU/mL) in DPBS without Ca^2+^ and Mg^2+^ using a separate vial. In another vial, human thrombin (500 IU/mg) was dissolved in calcium chloride solution (6 mg/mL). After mixing dissolved components from two vials, HCF hydrogel was gelated for 3 min at room temperature. Appendix A shows the gross appearance of gelated HCF hydrogel.

### 3.2. In Vitro Growth Factors Release

Targeted delivery and controlled release of growth factors or therapeutic proteins is one of the key tissue-engineering strategies to promote repair and regeneration of damaged tissues [38,39]. In order to examine whether HCF hydrogel is able to control the release of incorporated TGF-β1 and BMP-4 in the long term, we performed an in vitro growth factor release kinetics study. The release kinetics of TGF-β1 and BMP-4 incorporated within HCF hydrogel was determined by ELISA. Figure 2 shows that the release ratio of TGF-β1 and BMP-4 from the fibrin hydrogel was faster than that of HCF hydrogel.

The cumulative release of TGF-β1 from fibrin hydrogel and HCF hydrogel after 10 days was 94.4 ± 4.53% and 66.7 ± 3.76%, respectively (Figure 2A). At the same time interval, the release ratio of BMP-4 from fibrin hydrogel and HCF hydrogel achieved 92.4 ± 4.46% and 68.2 ± 4.12% (Figure 2B). An almost complete release of TGF-β1 and BMP-4 from HCF hydrogels was observed by the fourth week.

Thus, these results confirmed that HCF hydrogels possess properties on controlling the slow release of TGF-β1 and BMP-4 in a long-term manner, which would effectively be used as injectable system for the sustained delivery of two growth factors to achieve the long-term repair and regeneration of damaged cartilage tissue.

### 3.3. In Vitro Degradation Behavior of HCF Hydrogel

To examine the degradation behavior of the hydrogels, enzyme-mediated degradation and cell-mediated degradation tests were performed. As shown in Figure 3A, HCF hydrogels demonstrated a slower degradation rate in contrast to fibrin hydrogels. After incubation for 14 days, the weight loss of HCF hydrogel and HCF hydrogel containing TGF-β1 and BMP-4 (57.21 ± 3.12 and 59.17 ± 3.26%) was lower than that of fibrin hydrogel and fibrin hydrogel containing TGF-β1 and BMP-4 (51.35 ± 2.69 and 50.42 ± 2.52%). The incorporation of TGF-β1 and BMP-4 into hydrogels did not have effect on their degradation rate.

The cell-mediated degradation test revealed that HCF hydrogel had a slower degradation rate of weight loss than fibrin hydrogel. The final weights of gels counted from the initial hydrogel weight remained at 81.42% ± 4.91% and 75.27% ± 3.85%, respectively, over 14 days (Figure 3B). Thus, these results indicate that the incorporation of heparin-conjugated fibrinogen in hydrogel enhances the structural stability of HCF hydrogel.

### 3.4. Characterization of Rabbit SDMSCs

A number of studies have shown that the synovial membrane is a promising source of MSCs that can be used to repair and regenerate damaged articular cartilage [40,41]. SDMSCs in sufficient amounts can be easily isolated from a small piece of synovial membrane; they also do not lose their phenotypic properties during cultivation. At the same time, the functional activity of SDMSCs remains at a high level regardless of the age of the person [12]. Moreover, SDMSCs have been shown to have higher capacity to differentiate into chondrocytes than MSCs from bone marrow or adipose tissue [42]. It was also found that after injury, MSCs can migrate from the synovial membrane to the wound site and participate in the process of cartilage regeneration [43,44].

In this regard, in our study, we isolated and characterized rabbit SDMSCs to use them further in our in vitro and in vivo experiments. The synovium-derived cells were expanded up to passage 4 and characterized with phase-contrast microscopy, CFU-F assay and trilineage differentiation assay. As shown in Appendix A, rabbit SDMSCs exhibited homogeneous fibroblast-like morphology (Appendix A) and showed the capacity to form cell colonies (Appendix A). In addition, SDMSCs were capable of differentiating into adipocytes, chondrocytes and osteoblasts (Appendix A). Thus, these results indicate that obtained SDMSCs were suitable for in vitro and in vivo studies because of their clonogenic capacity and multilineage differentiation potential.

### 3.5. Proliferation and Viability of SDMSCs

It is known that fibrin is a naturally derived scaffold that possesses good biocompatibility properties with low cytotoxicity [45]. However, after fabrication of heparin-conjugated fibrinogen, toxic substances such as acetone might remain in the scaffold as a result of insufficient washing. In order to examine biocompatibility, rabbit SDMSCs were encapsulated in HCF hydrogel and incubated in complete culture medium α-MEM. The cell viability of SDMSCs encapsulated in the HCF hydrogel was confirmed by the Live/Dead cell viability assay at day 1, 3 and 7 after cultivation. As shown in Figure 4C,D, high cell viability (97%) of SDMSCs was observed for 7 days after encapsulation, indicating the biocompatibility and nontoxicity of HCF hydrogel. In addition, proliferation assay showed that SDMSCs were able to normally proliferate in HCF hydrogel. As shown in Figure 4B, the total DNA content significantly increased from day 1 to day 7, indicating that rabbit SDMSCs in HCF hydrogel had a high proliferative activity.

Thus, these data confirm that HCF hydrogel possesses good biocompatibility and can be suitable for in vivo implantation in osteochondral defect regeneration.

### 3.6. Macroscopic Evaluation of Cartilage Repair

In order to evaluate the regenerative properties of HCF hydrogels containing growth factors and/or SDMSCs, we implanted HCF hydrogels into a rabbit osteochondral defect model. The gross appearance images of osteochondral defect regeneration 12 weeks after surgery are presented in Figure 5A.

In the hydrogel group, approximately 25% of the defect area was covered by fibrous-like tissue. A central area of the defect in the hydrogel group was dark brown. In the hydrogel/TGF-β1/BMP-4 group, the defect area was covered by cartilage-like tissue with an incomplete integration with surrounding cartilage. In the hydrogel/MSC group, white fibrous-like tissue with small fissures covered approximately 100% of the defect area. In the hydrogel/MSC/TGF-β1/BMP-4 group, the defect area was completely filled by cartilage-like tissue with smooth surface.

Thus, overall repair assessment indicated that the total score for the hydrogel/MSC/TGF-β1/BMP-4 group (11.72 ± 1.15) was significantly higher than scores for the hydrogel/TGF-β1/BMP-4 (6.50 ± 0.82), hydrogel/MSC (9.61 ± 0.97) and hydrogel alone groups (1.53 ± 0.54) (Figure 5B).

### 3.7. Histological Evaluation of Cartilage Repair

In order to evaluate the quality of osteochondral defect repair, the paraffin-embedded specimens were cut into 5 µm sections and stained with H&E, Safranin O and collagen II immunocytochemistry. Representative histological images of the defect area of each group are shown in Figure 6. The defect area in the hydrogel-only group was not replaced with cartilage and subchondral bone tissue. High magnification of the histological sections revealed that the defects were filled with nonfunctional and disorganized tissue with negative Safranin O staining.

In the hydrogel/TGF-β1/BMP-4 group, the defects were repaired by a relatively thin layer of cartilage-like tissue with weak Safranin O staining of the cartilage ECM. Despite incomplete formation of cartilage-like tissue, the subchondral bone plate was well reconstituted at week 12 after the surgery.

The hydrogel/MSC group showed incomplete regeneration of articular cartilage and subchondral bone in the defect area. The defect area was filled with a mixed type of cartilage that consisted of fibrous, fibrocartilaginous and hyaline-like tissue. The repaired tissue exhibited irregular surface with shallow bend in the middle area of the defect. Fibrous and fibrocartilaginous tissue were observed in the superficial and middle layers of the repaired tissue. Hyaline-like cartilage was seen under the fibrous tissue and the deeper layer of the osteochondral defect. The cartilage ECM of the repaired tissue in the hydrogel/MSC group was unevenly stained with Safranin O. In the superficial layer, the repaired tissue showed weak Safranin O staining, whereas in deeper layers the cartilage ECM demonstrated moderate and strong Safranin O staining (Figure 6B).

Co-delivery of SDMSCs with TGF-β1 and BMP-4 in HCF hydrogel showed the highest regenerative capacity in rabbit osteochondral defect model. In the hydrogel/MSC/TGF-β1/BMP-4 group, the defect area was completely repaired by hyaline cartilage tissue and the subchondral bone plate by week 12 after the surgery. The repaired tissue had smooth surface and a good integration with the adjacent cartilage. Higher magnification of the repaired tissue showed that the cells had a typical columnar alignment. The morphology of cartilage cells appeared to resemble that of normal chondrocytes. Moreover, histochemical analysis showed that the cartilage ECM was normally and uniformly stained with Safranin O (Figure 6B).

In addition to histochemical analysis, we performed immunohistochemical staining for collagen II to completely evaluate the quality of regenerated articular cartilage. The results of immunohistochemical analysis are shown in Figure 6C. In the hydrogel group, the signs of cartilage repair and collagen II expression were not observed. The hydrogel/TGF-β1/BMP-4 group showed uneven collagen II immunostaining in the cartilage ECM. The hydrogel/MSC group showed that the superficial layer of the repaired cartilage tissue did not express collagen II. However, abundant collagen II expression was detected in the middle and deeper layers of the repaired tissue. The hydrogel/MSC/TGF-β1/BMP-4 group demonstrated the best regenerative effect in comparison with other groups. Expression of collagen II in the cartilage ECM of the hydrogel/MSC/TGF-β1/BMP-4 group was uniform and strong, indicating the presence of hyaline cartilage.

Moreover, as shown in Figure 7, histological scoring for osteochondral defects revealed that the total score for the hydrogel/MSC/TGF-β1/BMP-4 group (27.12 ± 1.87) was significantly higher than the hydrogel/MSC (17.05 ± 1.68), hydrogel/TGF-β1/BMP-4 (16.03 ± 1.45) and hydrogel-alone (2.10 ± 0.97) groups.

## 4. Discussion

In the field of regenerative medicine and tissue-engineering techniques, scaffolds, cells and growth factors are the main components for cartilage and osteochondral lesions treatment [8]. Currently, one of the attractive tissue-engineering approaches to promote the efficiency of articular cartilage therapy is co-delivery of MSCs and specific growth factors in hydrogels. Recently, a number of studies demonstrated that MSCs encapsulated in different hydrogels with chondroinductive growth factors significantly repaired cartilage and osteochondral defects in contrast to the individual application of MSCs or growth factors [46]. For example, Gugjoo et al. showed that the implantation of laminin gel-based scaffold containing MSCs with IGF-1 and TGF-β1 significantly promoted regeneration of damaged hyaline cartilage and subchondral bone tissue in a rabbit osteochondral model [47]. Rabbit adipose-derived MSCs encapsulated in poly-(d,l-lactic acid-co-glycolic acid) (PLGA) hydrogel with BMP-2 significantly improved cartilage repair in chondral defects compared to untreated control and microfracture treatment alone [48]. Kim et al. demonstrated that co-delivery of rabbit synovium-resident MSCs with BMP-7 via fibrous PLGA scaffolds had a synergistic therapeutic effect on osteochondral defect repair, resulting in complete hyaline cartilage and subchondral bone tissue regeneration [49]. In this study, we fabricated HCF hydrogel with SDMSCs and two growth factors (TGF-β1 and BMP-4), which play a key role in the induction of stem cell recruitment and chondrogenic differentiation. We further examined whether the combined delivery of SDMSCs, TGF-β1 and BMP-4 in HCF hydrogel can strengthen the effect of osteochondral defect repair in rabbit knee joints compared to the individual delivery of SDMSCs and growth factors.

One of the crucial factors in this approach is the slow release of growth factors from scaffolds to the defect area. In this regard, the uncontrolled burst release of growth factors or drugs from implanted hydrogels at the injured area remains one of the main clinical problems in regenerative medicine and targeted drug delivery. Heparin-functionalized hydrogels are considered as appropriate polymer carriers for controlled and sustained drug delivery—in particular, heparin-binding growth factors such as TGF- βs and BMPs [28]. Heparin is a sulfated glycosaminoglycan, which can interact with various proteins and growth factors, forming stable complexes to regenerate damaged tissues [50]. In our study, the choice of heparin-functionalized hydrogel is justified by several advantages: (1) the ability to protect growth factors from the action of proteases and to maintain their bioactivity for a long time in vivo; (2) sustained release of growth factors from the hydrogel provided by electrostatic interaction of negatively charged heparin with electropositive proteins.

Considering the aforementioned advantages, in our study, we simultaneously incorporated TGF-β1 and BMP-4 in HCF hydrogel and found that the sustained release of these growth factors from the hydrogel lasts up to 4 weeks. Moreover, an in vitro degradation test showed that HCF hydrogel demonstrated a slower degradation rate in contrast to normal fibrin hydrogel. These data are in good consistency with previous study by Yang et al., which showed that in vivo biodegradation of HCF hydrogel was significantly slower than that of fibrin hydrogel for 9 days after implantation to the subcutaneous layer of the athymic mice [51]. Thus, these data indicated that the incorporation of heparin-conjugated fibrinogen compared to fibrin hydrogel had a significant effect on the degradation behavior and release rate of TGF-β1 and BMP-4 for a long period of time. In addition, our in vitro study also demonstrated that HCF hydrogel exhibited high biocompatibility and supported the proliferation of SDMSCs, as demonstrated by the Live/Dead cell viability assay and DNA content assay.

Microscopic analysis showed that incorporated SDMSCs were evenly distributed in HCF hydrogel and the cells mostly appeared as round-shaped for 7 days of cultivation in the complete culture media. In our opinion, the reason for the rounded morphology of incorporated SDMSCs is associated with the high fibrinogen concentration in HCF hydrogel. In the present study, the final fibrinogen concentration in HCF hydrogel was approximately 45–50 mg/mL. According to literature data, MSCs loaded in fibrin hydrogel with low fibrinogen concentrations (5 mg/mL) had a spindle-like shape, but at high fibrinogen concentrations (45–50 mg/mL), MSCs maintained a mostly rounded morphology [52]. Despite the high concentration of fibrinogen, MSCs in fibrin hydrogel sustain functional survival and paracrine function through the production of growth factors and immunomodulatory mediators [53].

In order to evaluate individual and synergistic therapeutic effects of SDMSCs and growth factors, we implanted the following HCF hydrogels into the osteochondral defects of rabbits: (1) HCF hydrogel alone; (2) HCF hydrogel containing TGF-β1 and BMP-4; (3) HCF hydrogel containing SDMSCs; and (4) HCF hydrogel containing SDMSCs, TGF-β1 and BMP-4. Macroscopic and histological evaluations showed that HCF hydrogel alone was not capable of repairing cartilage and subchondral bone tissue in osteochondral defects within 12 weeks. At the same time, the implantation of HCF hydrogel with SDMSCs resulted in the formation of a mixed type of cartilage instead of hyaline tissue. The reasons for this failure may be associated with an insufficient concentration of specific growth factors required for recruitment and chondrogenic differentiation of MSCs in the defect area [54].

Another key factor in this tissue-engineering technique is the selection of the appropriate growth factors. A number of studies showed that growth factors of the TGF-β superfamily play an important role in cartilage tissue repair. TGF-βs and BMPs have the ability not only to effectively stimulate the matrix formation and chondrogenic differentiation of MSCs, but also can recruit endogenous progenitor cells to the damaged area to promote cartilage tissue regeneration [55,56,57]. Indeed, our animal study demonstrated that the implantation of HCF hydrogel with TGF-β1 and BMP-4 significantly promoted the regeneration of osteochondral defects. In this case, the regeneration of osteochondral defects may be attributed to chondrogenic and osteogenic differentiation of the recruited endogenous MSCs by long-term delivery of TGF-β1 and BMP-4. However, even the subchondral bone tissue was well reconstructed and the newly formed cartilage layer was thin and weakly stained with Safranin O, indicating low proteoglycan content in the cartilage ECM. Apparently, incomplete cartilage tissue repair was due to insufficient recruitment of endogenous MSCs and chondroprogenitor cells.

An enhanced therapeutic effect was found in the animal group when we used the combined delivery of SDMSCs and chondrogenic growth factors (TGF-β1 and BMP-4) for regeneration of osteochondral defects. In this group, the defect area was completely repaired by hyaline cartilage and subchondral bone tissue. Moreover, the hydrogel/MSC/TGF-β1/BMP-4 group demonstrated the best cartilage integrity and the greatest cartilage ECM deposition when evaluated by H&E, Safranin O and collagen II immunocytochemical staining. Histological scores also were the highest for the hydrogel/MSC/TGF-β1/BMP-4 group among all experimental groups. Thus, these results suggested that combined delivery of SDMSCs and chondrogenic factors in HCF hydrogel may have a synergistic therapeutic effect in promoting osteochondral defect repair.

The mechanism for complete osteochondral defect repair with SDMSCs and chondrogenic factors might be explained by providing the biomimetic microenvironment required for the effective regeneration of hyaline cartilage and subchondral bone tissue by HCF hydrogel. Firstly, HCF hydrogel after implantation into osteochondral defects provides sustained releases of TGF-β1 and BMP-4, and thereby can recruit endogenous bone marrow MSCs and other progenitor cells to the defect area for tissue repair. Secondly, in the presence of TGF-β1 and BMP-4, recruited endogenous MSCs and exogeneous MSCs effectively differentiate into mature chondrocytes, accompanied by a production of cartilage ECM proteins such as proteoglycans, glycoaminoglycans and collagen II. It was confirmed earlier by Nakayama et al., who showed that exogenously added BMP-4 acts synergistically with TGF-β3 in inducing chondrogenesis in MSCs and chondroprogenitor cells [58]. In addition, it was found that both TGF-β3 and BMP-4 are required for stimulation of chondrogenic differentiation of MSCs, while chondroprogenitor cells can differentiate into mature chondrocytes in the presence of BMP-4 only. Thirdly, the anti-inflammatory properties of SDMSCs may be involved in the mechanism of cartilage tissue repair. It has been demonstrated that human SDMSCs can significantly suppress proliferation of CD4^+^ T cells stimulated by CD3/CD28 and reduce the levels of TNF-α, IFN-γ and IL-17A in experimental animals [59]. Kim et al. showed that MSCs loaded in fibrin gel effectively inhibited immune reactions through the secretion of growth factors and immunomodulatory factors such as TGF-β1, HGF and PGE_2_ [53]. Thus, we suppose that the cartilage repair mechanism in our osteochondral defect model is associated with the recruitment and chondrogenic differentiation of endogenous and exogeneous MSCs due to long-term co-delivery of TGF-β1 and BMP-4 in the defect area and immunomodulatory effects of SDMSCs.

## 5. Conclusions

In conclusion, this study demonstrated that HCF hydrogel exhibited a good level of biocompatibility for encapsulated SDMSCs and was able to control the release of TGF-β1 and BMP-4 in the long term. An in vivo study revealed that the implantation of autologous SDMSCs in HCF hydrogel in combination with TGF-β1 and BMP-4 significantly enhanced the regeneration of osteochondral defects in rabbits through the complete formation of hyaline cartilage and subchondral bone tissue compared to HCF hydrogels with SDMSCs or growth factors alone. These data suggest that the combined delivery of SDMSCs with TGF-β1 and BMP-4 in HCF hydrogel might serve as a promising tool in tissue-engineering strategies for repairing osteochondral articular defects.

## Figures and Tables

**Figure 1 polymers-14-05343-f001:**
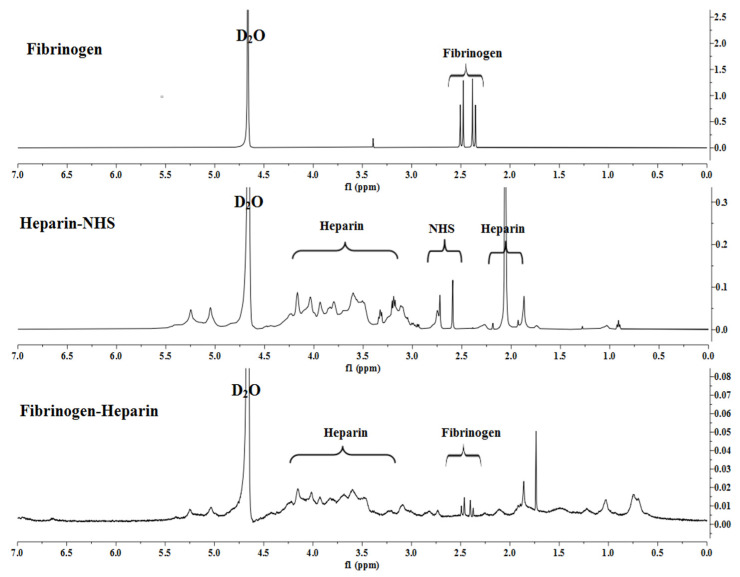
^1^H-NMR spectra of fibrinogen, heparin and heparin-conjugated fibrinogen.

**Figure 2 polymers-14-05343-f002:**
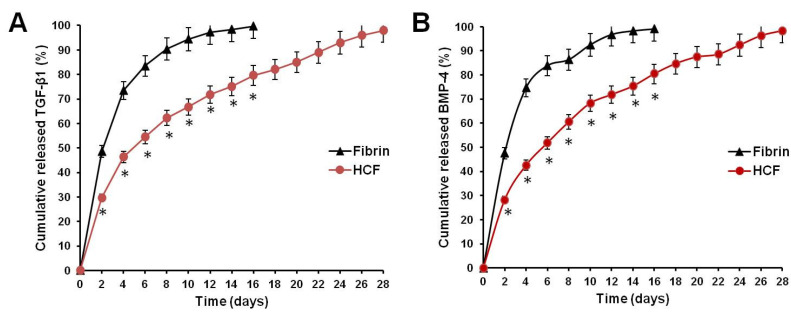
Profiles of cumulative release of TGF-β1 (**A**) and BMP-4 (**B**) from fibrin and HCF hydrogels. The values represent mean ± standard deviation (*n* = 6). * Significant difference from fibrin group, *p* ≤ 0.05.

**Figure 3 polymers-14-05343-f003:**
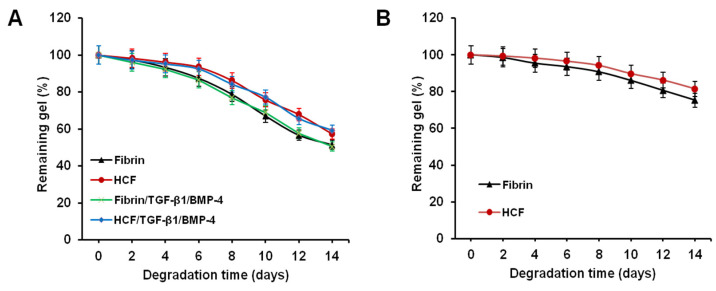
In vitro degradation kinetics of fibrin and HCF hydrogels. (**A**) Enzyme-mediated degradation of the hydrogels. (**B**) Cell-mediated degradation of the hydrogels.

**Figure 4 polymers-14-05343-f004:**
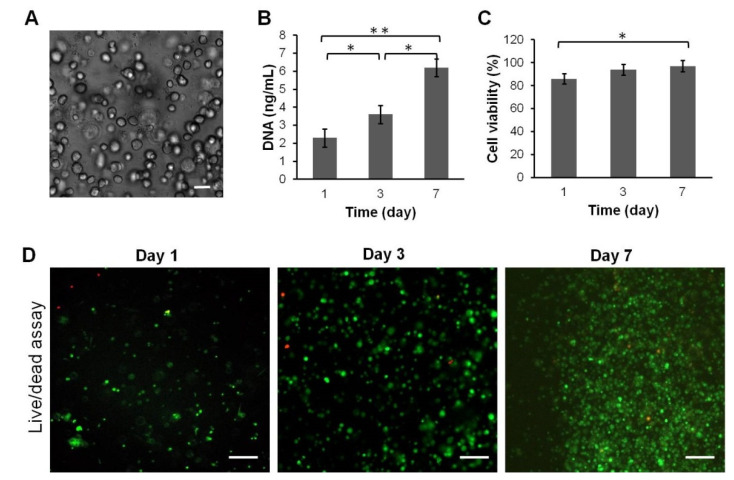
Cell viability and proliferation of rabbit SDMSCs encapsulated in HCF hydrogel. (**A**) Representative image of SDMSCs encapsulated in HCF hydrogel. Scale bar 20 µm. (**B**) Proliferation of SDMSCs encapsulated in HCF hydrogel. (**C**) Cell viability percentage of SDMSCs from the total cell number *(n* = 3). (**D**) Fluorescent images of Live/Dead staining of SDMSCs encapsulated in HCF hydrogel. Live cells (Green) and dead cells (Red). Scale bar 100 µm. * *p* < 0.05, ** *p* < 0.01.

**Figure 5 polymers-14-05343-f005:**
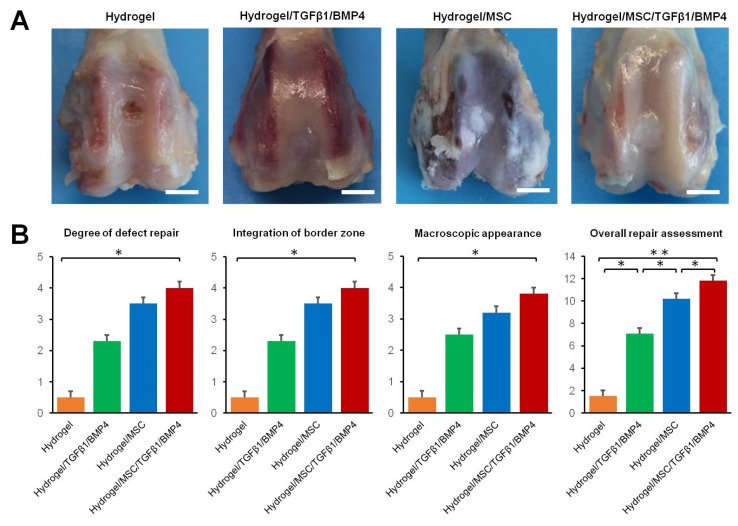
Macroscopic evaluation for osteochondral defect repair. (**A**) Macroscopic appearance of the defects at week 12 after implantation of hydrogels. Scale bar 4 mm. (**B**) ICRS scores of repaired osteochondral defects at week 12. Data are presented as mean ± SD (*n* = 4). * *p* < 0.05, ** *p* < 0.01.

**Figure 6 polymers-14-05343-f006:**
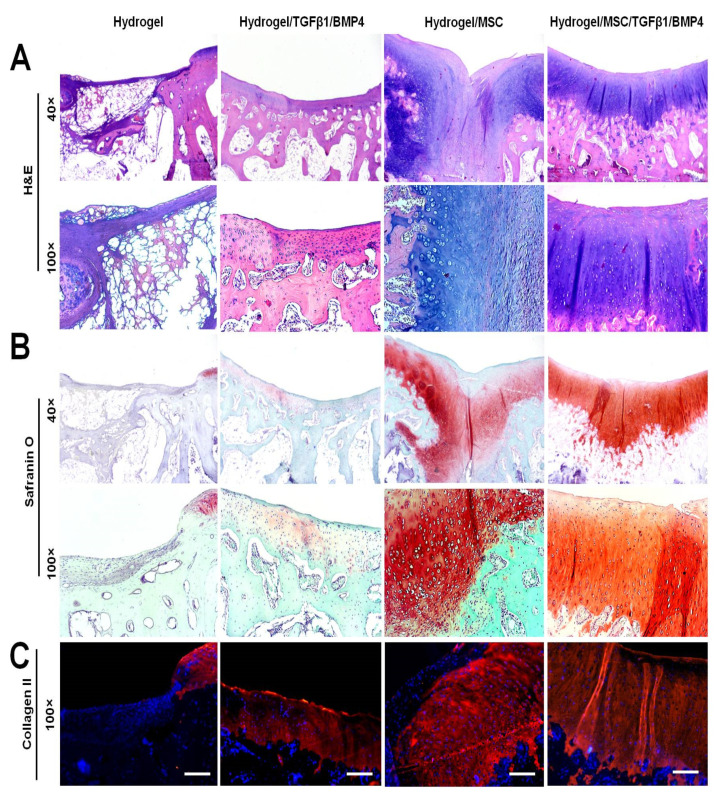
Histological analysis of the osteochondral defect repair at week 12 after hydrogel implantation. (**A**) H&E staining; (**B**) Safranin O staining; (**C**) Immunohistochemical staining of collagen II (red). Cell nuclei are stained with DAPI (blue). Scale bar, 100 µm.

**Figure 7 polymers-14-05343-f007:**
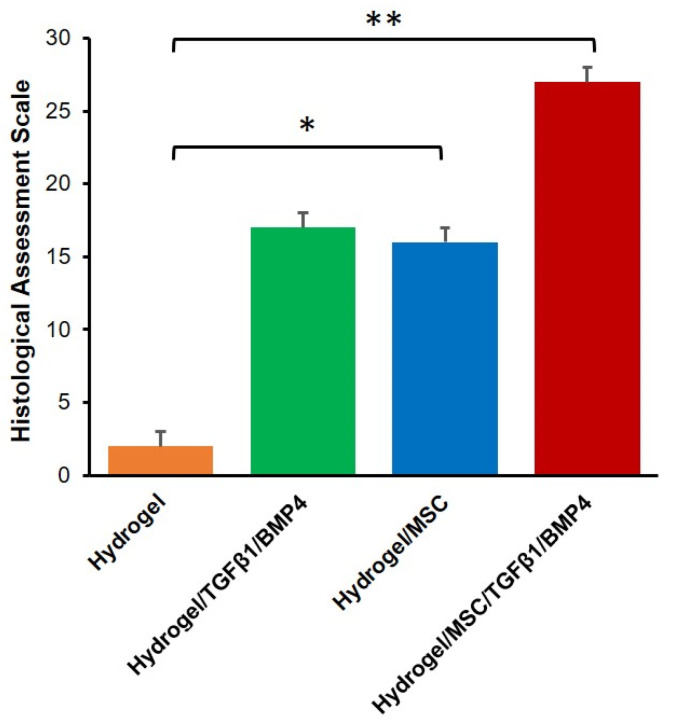
Histological assessment scale for repaired osteochondral defects. Data are presented as mean ± SD (n = 4). * *p* < 0.05, ** *p* < 0.01.

**Table 1 polymers-14-05343-t001:** Experimental groups of rabbits for hydrogel implantation.

#	Group	Description
(i)	Hydrogel (negative control)	Administration of HCF hydrogel only
(ii)	Hydrogel/MSC	Administration of HCF hydrogel with autologous SDMSCs (1 × 10^6^ cells/70 µL)
(iii)	Hydrogel/TGF-β1/BMP-4	Administration of HCF hydrogel with TGF-β1 (200 ng/70 µL) and BMP-4 (200 ng/70 µL)
(iv)	Hydrogel/MSC/ TGF-β1/BMP-4	Administration of HCF hydrogel with autologous SDMSCs (1 × 10^6^ cells/70 µL), TGF-β1 (200 ng/70 µL) and BMP-4 (200 ng/70 µL)

## Data Availability

The data presented in this study are available on request from the corresponding author.

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
