# Peer review of "Regeneration of Osteochondral Defects by Combined Delivery of Synovium-Derived Mesenchymal Stem Cells, TGF-β1 and BMP-4 in Heparin-Conjugated Fibrin Hydrogel"

_polymers, 2022, doi:10.3390/polym14245343_

Round 1
Reviewer 1 Report
Sarsenova and Ogay et al. fabricated a heparin-conjugated fibrin hydrogel with synovium-derived mesenchymal stem cells, transforming TGF-β1 and BMP-4 to repair osteochondral defects in a rabbit model. The results showed that the hydrogel exhibited a high level of biocompatibility and was able to control the release of TGF-β1 and BMP-4. In vivo study revealed that implantation of the hydrogel with cells, TGF-β1 and BMP-4 enhanced regeneration of hyaline cartilage and subchondral bone plate in osteochondral defect.
Some additional comments are given below:
1. There are two Figure 2 in the manuscript. The author pays attention to the serial number of the figures.
2. The H&E staining in Figure 7 is not sufficient for tissue repair. It is recommended that the authors evaluate the expression of common cartilage-related proteins by immunohistochemical methods.
3. The backgrounds in the manuscript images are mostly different, and the authors should make adjustments.
Author Response
- There are two Figure 2 in the manuscript. The author pays attention to the serial number of the figures.
- Thank you for the comment! We have corrected the number of the figures.
- The H&E staining in Figure 7 is not sufficient for tissue repair. It is recommended that the authors evaluate the expression of common cartilage-related proteins by immunohistochemical methods.
- Thank you for the recommendation! We purchased anti-collagen type 2 from Abcam (UK) and conducted immunohistochemical staining. The results are presented in Figure 6.
- The backgrounds in the manuscript images are mostly different, and the authors should make adjustments.
- Thank you for the comment! We have adjusted the backgrounds in the images.
Reviewer 2 Report
There are no new ideas in the manuscript. Synovium membrane-derived stem cells are well known in cartilage tissue engineering, and the immobilization of heparin-conjugated hydrogels has also reported as a functional scaffold for growth factor immobilization in tissue engineering. In addition, TGF and BMP have been widely used for the bone and cartilage tissue regeneration.
Figure 1 (total amount in vivo), Figure 2 (gelatin image) and Figure 4 (SDMC characterization) should be moved to supplementary information and the authors should present the underlying mechanism of hydrogel/MSC/TGF/BMP compared to the hydrogel group and the hydrogel/TGF/BMP group.
In my opinion on Figure 7, the Hydrogel group (D-F) exhibits adequate regenerative properties without fracture-related trabecular thickening.
In addition, in order to evaluate ECM and GAG in regenerated cartilage, toluidine blue or Safranin O staining results should be presented.
Author Response
There are no new ideas in the manuscript. Synovium membrane-derived stem cells are well known in cartilage tissue engineering, and the immobilization of heparin-conjugated hydrogels has also reported as a functional scaffold for growth factor immobilization in tissue engineering. In addition, TGF and BMP have been widely used for the bone and the authors should present the underlying mechanism of hydrogel/MSC/TGF/BMP compared to the hydrogel group and the hydrogel/TGF/BMP group.
- Thank you for the comment! We have moved Fig 1, Fig 2 and Fig. 4 to the supplementary materials. The underlying mechanism of hydrogel/MSC/TGF/BMP compared to the hydrogel group and the hydrogel/TGF/BMP group we describe in section “Discussion”.
In addition, in order to evaluate ECM and GAG in regenerated cartilage, toluidine blue or Safranin O staining results should be presented.
-Thank you for the recommendation! To evaluate proteoglycan and GAG content in cartilage tissue we have conducted additional histochemical staining by Safranin O. The results are presented in Figure 6.
Reviewer 3 Report
In this manuscript entitled “Regeneration of Osteochondral Defects by Combined Delivery of Synovium-Derived Mesenchymal Stem Cells, TGF-β1 and 3 BMP-4 in Heparin-Conjugated Fibrin Hydrogel”, the authors present data demonstrate that HCF hydrogel incorporating with MSCs and growth factors showed the regeneration of cartilage and osteochondral defects in a rabbit model. The reviewer agrees with the concepts. Even though the results are interesting, the presentation of statistical data is inadequate. I have a few comments, explained below. The following points should be addressed for paper re-submission.
Major comments
1) Introduction: The authors explain the advantages of synovium-derived MSCs (SDMSCs) compared with MSCs isolated from other tissues. However, many stem cell researchers understand that umbilical cord-derived MSCs have high therapeutic potential. A More convincing Introduction should be presented.
2) Figure 3: Release profiles of growth factors are shown. It is also important to study the degradation behavior of hydrogels.
3) Figure 5B and C: Statistical analysis is required.
4) Figure 6B: The statistical data is inadequate.
5) Figure 8: Statistical analysis is required. Is the effect of treatment with Hydrogel/MSC/TGFb1/BMP4 additive or synergistic?
6) Does this hydrogel exhibit inflammatory properties locally at the implantation site? If there are sufficient tissue sections remaining, it is recommended that this evaluation be added.
Minor comments
1) Results and Discussion: It is recommended to separate “Results” and ”Discussion”. It would help the readers to understand.
Author Response
- Introduction: The authors explain the advantages of synovium-derived MSCs (SDMSCs) compared with MSCs isolated from other tissues. However, many stem cell researchers understand that umbilical cord-derived MSCs have high therapeutic potential. A More convincing Introduction should be presented.
- Thank you for the comment! In our paper we focused on synovium-derived MSCs (SDMSCs) and their properties. SDMSCs are tissue-resident stem cells and they have the highest chondrogenic potential compared to MSCs isolated from bone marrow, adipose tissue and umbilical cord. In this regard, SDMSCs are one of the best cell sources for cartilage regeneration. We think that the introductory part of our manuscript is sufficiently described for readers.
2) Figure 3: Release profiles of growth factors are shown. It is also important to study the degradation behavior of hydrogels.
- Thank you for the recommendation! We have conducted additional study on evaluation of degradation property of HCF hydrogel using enzyme-mediated degradation test and cell-mediated degradation test. The results are presented in Figure 3.
3) Figure 5B and C: Statistical analysis is required.
- Thank you for the comment! We have corrected the statistics
4) Figure 6B: The statistical data is inadequate.
- Thank you for the comment! We have corrected the statistics
5) Figure 8: Statistical analysis is required. Is the effect of treatment with Hydrogel/MSC/TGFb1/BMP4 additive or synergistic?
- Thank you for the comment! We think that combined delivery of SDMSCs and chondrogenic growth factors may synergistically promote cartilage tissue repair.
6) Does this hydrogel exhibit inflammatory properties locally at the implantation site? If there are sufficient tissue sections remaining, it is recommended that this evaluation be added.
- Thank you for comment! Unfortunately, we have not sufficient tissue sections for this evaluation.
Minor comments
Results and Discussion: It is recommended to separate “Results” and ”Discussion”. It would help the readers to understand.
- Thank you for comment! We have separated the sections “Results” and ”Discussion”.
Reviewer 4 Report
The article "Regeneration of Osteochondral Defects by Combined Delivery 2 of Synovium-Derived Mesenchymal Stem Cells, TGF-β1 and 3 BMP-4 in Heparin-Conjugated Fibrin Hydrogel" is a routine, qualitative presentation of the results of an in vitro/in vivo experiment on the restoration of cartilage and bone tissue in a model of bone and cartilage defect. Both the proposed methodology, the prototype of the product, and the design of the experiment are traditional for regenerative medicine.
The design of the study, the methods used to evaluate the result are generally quite adequate; however, the authors should be invited to “upgrade” the text for the audience to be comfortable understanding the results.
1. The design of the experiment on animals is not quite clear; it is advisable to clarify why a group with independent healing of the defect was not introduced into the work.
2. Statistical processing needs to be improved, both from the point of view of the applied semi-quantitative analysis and from the point of view of graphical display.
3. Histological methods of investigation: the above photo of chondrospheres does not make it possible to verify the synthesis of GAG; a fragment is needed at a higher magnification; preparations of cartilage and bone defects are generally informative; if possible, micrographs should be edited by color-tone correction, and it is also necessary to filter dyes to avoid their particles falling out on the preparation, which litters micrographs.
4. Staining with hematoxylin and eosin gives an idea of the histoarchitecture of the regenerate tissues formed and allows to identify bone and cartilage tissues, and among them to diagnose subtypes: reticulofibrous and lamellar bone tissue; fibrous and hyaline cartilage tissue. Of course, additional coloring methods can clarify the features of the biochemistry of the matrix, but they cannot radically affect the diagnosis of the type of tissue, which allows them to be used optionally.
5. It would be correct to suggest to the authors in Fig. 1 to cut off the sections of the photo with wool; firstly, these sections are not informative; secondly, they arouse fears of unsterile surgery among reviewers and readers.
The authors should be invited to expand the "discussion" section. It would be useful to know their opinion regarding the fact that, judging by the photo at low magnification, even after 7 days in the gel, the cells did not form processes, but remained spherical.
Due to what the revealed differences in the composition of regenerates were formed? Still, the reference to simple synergy needs a more detailed discussion.
General conclusion: I consider, that the article, despite its moderate novelty, can be edited by the authors taking into account the comments of reviewers and re-submitted to a special issue for publication.
Author Response
The design of the study, the methods used to evaluate the result are generally quite adequate; however, the authors should be invited to “upgrade” the text for the audience to be comfortable understanding the results.
- The design of the experiment on animals is not quite clear; it is advisable to clarify why a group with independent healing of the defect was not introduced into the work.
- The hydrogel group was used as a negative control, which originally did not result in the desired outcome of the experiment. This group was used to ensure that there is no response to the treatment and helped to identify the influence of external factors such as MSCs and growth factors. Hydrogel group as an independent variable allowed to determine that the outcome was caused by the experimental treatment and not by other variables. Thus, single hydrogel implantation served as a baseline for determining the effectiveness of the treatment with growth factors and MSCs separately or in combination and, therefore, there it was not necessary to use untreated control group.
- Additionally, during the development of the experimental design, we referred to the studies which were published earlier and the authors did not used untreated control groups [Lee, J.M.; Im, G.I. SOX trio-co-transduced adipose stem cells in fibrin gel to enhance cartilage repair and delay the progression of osteoarthritis in the rat. Biomaterials 2012, 33, 2016–2024; Lee JM, Kim BS, Lee H, Im GI. In vivo tracking of mesechymal stem cells using fluorescent nanoparticles in an osteochondral repair model. Mol Ther. 2012 Jul;20(7):1434-42].
- Statistical processing needs to be improved, both from the point of view of the applied semi-quantitative analysis and from the point of view of graphical display.
- Thank you for the comment! We have corrected the statistics.
- Histological methods of investigation: the above photo of chondrospheres does not make it possible to verify the synthesis of GAG; a fragment is needed at a higher magnification; preparations of cartilage and bone defects are generally informative; if possible, micrographs should be edited by color-tone correction, and it is also necessary to filter dyes to avoid their particles falling out on the preparation, which litters micrographs.
- Thank you for the comment! We have edited our images and improved their quality by ImageJ software.
- Staining with hematoxylin and eosin gives an idea of the histoarchitecture of the regenerate tissues formed and allows to identify bone and cartilage tissues, and among them to diagnose subtypes: reticulofibrous and lamellar bone tissue; fibrous and hyaline cartilage tissue. Of course, additional coloring methods can clarify the features of the biochemistry of the matrix, but they cannot radically affect the diagnosis of the type of tissue, which allows them to be used optionally.
- Thank you for the comment! To evaluate collagen 2 expression, proteoglycan and GAG content in cartilage tissue we have conducted additional staining by Safranin O and anti-collagen 2 (Abcam, UK). The results are presented in Figure 6.
- It would be correct to suggest to the authors in Fig. 1 to cut off the sections of the photo with wool; firstly, these sections are not informative; secondly, they arouse fears of unsterile surgery among reviewers and readers.
- We appreciate the comment. Recommendation for cutting of the parts with rabbit fur is valuable but cropping might impact on the readability and acceptance of the pictures by readers. This is also technically challenging and will worsen the presentability of the information.
The authors should be invited to expand the "discussion" section. It would be useful to know their opinion regarding the fact that, judging by the photo at low magnification, even after 7 days in the gel, the cells did not form processes, but remained spherical.
- Thank you for comment! We separated the sections “Results” and ”Discussion”.
Round 2
Reviewer 1 Report
Authors have improved the manuscript significantly in regards to the original version. From my point of view, the new version of the manuscript that is including the answers to my questions and inquiries has been well performed. For me, this new version is including all indicated aspects for considering to be integrated.
Reviewer 2 Report
I did not change my opinion on the manuscript.
The author has not yet presented data on the mechanism of action of the stem cell and growth factor complex in the revised paper. There is also a lack of in-depth review of revised papers and a lack of expertise in data representation.
1. In the case of a revised thesis, the results of M&M 2.3 and 2.4 have been deleted, and the contents of the M&M remain the same.
2. The contents of Table 1 overlap lines 232-239 and should be deleted.
3. This is the part where the results of Figure 2-3 should be described.
4. Figure 5B should be deleted as data that is not only subjective but lacks scientific evidence, and Figure 5A should be combined with Figure 6 and Figure 7 to explain the results.
Reviewer 3 Report
Accept in present form